# Dihydropyrazole Derivatives Containing Benzo Oxygen Heterocycle and Sulfonamide Moieties Selectively and Potently Inhibit COX-2: Design, Synthesis, and Anti-Colon Cancer Activity Evaluation

**DOI:** 10.3390/molecules24091685

**Published:** 2019-04-30

**Authors:** Xiao-Qiang Yan, Zhong-Chang Wang, Bo Zhang, Peng-Fei Qi, Gui-Gen Li, Hai-Liang Zhu

**Affiliations:** 1State Key Laboratory of Pharmaceutical Biotechnology, Nanjing University, Nanjing 210023, China; 18120176083@163.com (X.-Q.Y.); zhangbo_nju@163.com (B.Z.); qipengfeinju@163.com (P.-F.Q.); 2Jiangsu Key Laboratory of Advanced Organic Materials, School of Chemistry and Chemical Engineering, Nanjing University, Nanjing 210023, China

**Keywords:** dihydropyrazole derivatives, sulfonamide, COX-2 inhibitors, docking simulation, pharmacological efficiency, colon tumor therapeutics

## Abstract

Cyclooxygenase-2 (COX-2) as a rate-limiting metabolism enzyme of arachidonic acid has been found to be implicated in tumor occurrence, angiogenesis, metastasis as well as apoptosis inhibition, regarded as an attractive therapeutic target for cancer therapy. In our research, a series of dihydropyrazole derivatives containing benzo oxygen heterocycle and sulfonamide moieties were designed as highly potent and selective COX-2 inhibitors by computer-aided drug analysis of known COX-2 inhibitors. A total of 26 compounds were synthesized and evaluated COX-2 inhibition and pharmacological efficiency both in vitro and in vivo with multi-angle of view. Among them, compound **4b** exhibited most excellent anti-proliferation activities against SW620 cells with IC_50_ of 0.86 ± 0.02 µM than Celecoxib (IC_50_ = 1.29 ± 0.04 µM). The results favored our rational design intention and provides compound **4b** as an effective COX-2 inhibitor available for the development of colon tumor therapeutics.

## 1. Instruction

Cyclooxygenase (COX), also known as prostaglandin oxidase reductase, is a bifunctional enzyme with cyclooxygenase and catalase activities, and plays a key rate-limiting role in the conversion of arachidonic acid into prostaglandin. According to the current research, there are three isozymes for cox-oxidase, such as COX-1, COX-2, and COX-3 [1,2]. However, three isozyme types not only have great differences in intracellular quantity and distribution area, but also in physiological functions [3]. COX-1 is a structural primary enzyme, which mainly exists in gastrointestinal tract, kidney and other parts. Its function is to promote the synthesis of physiological prostaglandin and regulate the physiological activities of normal tissue cells by protecting the digestive tract mucosa or changing the vascular tension. COX-3 is a spliced variant of COX-1 isozyme, which is only found in the brain of dogs, but not in the human body [4,5]. Cox-2, by contrast, is an induced isozyme, of which the expressions are generally quite low in normal tissue cells [6]. Only when being in cancer cells or cells are stimulated by inflammatory factors, the expression level of COX-2 can be significantly increased, which can be increased to about 80 times of the normal level, leading to the increase of the content of PEG_2_, PGI_2_, and PGE_1_ in the inflammatory site, and then leading to inflammatory reaction and tissue damage [7,8]. The differences of COX-2 and other isozymes in structure, function, expression and distribution area lay a theoretical foundation for the design of selective COX-2 inhibitors, which will not affect the original physiological function of COX as much as possible [9,10,11]. In fact, COX-2 has long been used to treat pain and inflammation [12]. With the deepening and development of drugs in recent years, the research on selective COX-2 inhibitors as specific anti-tumor drugs is increasing dramatically [13,14]. Many studies have confirmed that the overexpression of COX-2 in various malignant tumor cells is closely related to the occurrence, growth, metastasis, and inhibition of apoptosis of cancer [15]. In other words, the selective COX-2 inhibitors as a targeted therapeutic drug can prevent the occurrence and growth of malignant tumors [16,17]. Considering the large gap between the supply and demand of anticancer drugs, it is of great significance to improve the availability of all kinds of anticancer drugs [18]. Therefore, this study attempts to design a class of selective COX-2 inhibitors and evaluate their potential in cancer drug development. 

According to the clinical trial database, hundreds of clinical trials have tested the anti-cancer potential of known COX-2 inhibitors such as Celecoxib [19,20,21]. Similar structure–activity relationships and modification strategies summarized by different known COX-2 inhibitors should be applicable to the design of new COX-2 inhibitors, which provides a design basis for the study of anticancer drugs based on cox-2 inhibitors [22,23]. Generally, diacyl heterocyclic with a five-membered core has been known as common COX-2 inhibitors drug skeletons to be widely used and studied [24,25,26]. In addition, *para*-sulfamoylphenyl moiety in sulfonamides plays a crucial role in the selection of COX-2, and many COX-2 inhibitors, such as Celecoxib, Valerian and Parecoxib [27,28,29]. (Figure 1) Besides, the dihydropyrazole skeleton also appeared frequently in our previously published studies on COX-2 inhibitors such as **A-16A**, **B-48**, and **C-4d** [30,31,32]. On this basis, we developed a new type of diaryl heterocyclic sulfanilamide scaffolds with dihydropyrazole as the five-membered core ring. The scaffold was replaced with a small molecular library where the molecules were sorted by docking score using virtual screening. Better active molecules were then synthesized and tested in vivo and in vitro to describe their pharmacological properties. The results suggest that we have identified a new class of COX-2 inhibitors available for the development of colon tumor therapeutics. 

## 2. Results and Discussion

### 2.1. Chemistry

The routes to synthesizing the novel dihydropyrazole derivatives containing benzo oxygen heterocycle and sulfonamide moieties **4a**–**4z** are outlined in Scheme 1. According to Experimental Section, the target compounds can be obtained through the three-step reaction. All synthesized compounds structures were exhibited in Table 1, which were reported for the first time and characterized by melting test, ^1^H NMR, ESI-MS and elemental analysis. The structural characterization parameters fully conform to the designed structure, and the related spectral data have been proposed in the Appendix A. Meanwhile, single crystal X-ray diffraction structure analysis further identified representative compound to determine the specific type of skeleton structure. These crystal data and corresponding Cambridge Crystallographic Data Centre (CCDC) number were presented in Figure 2 and Table 2, giving a perspective view of these compounds together with the atomic labeling system. 

### 2.2. Biological Activity

#### 2.2.1. Selective COX-2 Inhibitory Activities

Human COX-1/COX-2 ELISA kit was respectively, used to evaluate the COX-1 and COX-2 inhibitory of 26 synthetic dihydropyrazole derivatives containing benzo oxygen heterocycle and sulfonamide moieties **4a**–**4z**, with Celecoxib as a positive control drug. As shown in Table 3, most of the tested compounds showed prominent COX-2 inhibitory activity, while the inhibitory activity of these compounds on COX-1 was not obvious. This indicated that the series of compounds had better selectivity in inhibiting COX-2. Among them, compound **4b** showed the best inhibitory activity of COX-2 with IC_50_ values of 0.35 ± 0.02 μM, even better than the positive control drug Celecoxib (IC_50_ = 0.41 ± 0.03 μM). Besides, Compound **4b** and Celecoxib, exhibited an equal COX-2 selectivity index (IC_50, COX-1_/IC_50, COX-2_). Not only that, compound **4b** also exhibited an equal COX-2 selectivity index (IC_50, COX-1_/IC_50, COX-2_ = 137.3) compared with Celecoxib ((IC_50, COX-1_/IC_50, COX-2_ = 145.8). Furthermore, the compounds with 1,3-dioxolane **4a**–**4j** generally exhibited more potent anticancer activities than compounds having 1,4-dioxane **4k**–**4z**. It indicated benzo 1,3-dioxolane could benefit to enhance COX-2 inhibitory activities. 

#### 2.2.2. In Vitro Antiproliferative Activities

All synthetic dihydropyrazole derivatives containing benzo oxygen heterocycle and sulfonamide moieties **4a**–**4z** were evaluated for antiproliferation activities against 5 cancer cell lines (SW620, MCF-7, HeLa, A549 and HepG2) and one noncancer cell line, NCM460 by the MTT assay, in comparison with the reference Celecoxib. As shown in Table 4, the antiproliferative activities of the tested compounds were more pronounced against human colorectal carcinoma SW620 cells than the other cell lines. Compound **4b** exhibited the best antiproliferative activity among the tested compounds, with IC_50_ values of 0.86 ± 0.02 μM, especially against SW620 cells, compared with positive control drug Celecoxib (1.29 ± 0.04 μM). As for SW620, compounds with 1,3-dioxolane **4a**–**4j** behaved more potent antiproliferative activities than compounds with 1,4-dioxane **4k**–**4z**. Besides, the MTT assay was performed on a non-cancer line NCM460 to assess the cytotoxicity of the test compound to normal colon cells. The results showed that compound **4a**–**4z** and Celecoxib had similar low cytotoxic effect on NCM460, so these series of compounds had desirable safety. 

#### 2.2.3. Compound 4b Induced SW620 Cell Apoptosis

To confirm that antiproliferation activities of SW620 were associated with apoptosis, SW620 cell apoptosis induced by compound **4b** was determined using flow cytometry with Annexin V-FITC/PI Apoptosis Detection Kit. The outcome was shown in Figure 3, which indicated that the percentage of apoptotic cells was significantly increased in a dose-dependent manner. The percentages of cell apoptosis 7.9%, 19.9%, 45.9%, and 65.0% were responding to the concentration of compound **4b** 0 μM, 2.0 μM, 4.0 μM, and 8.0 μM, respectively. 

#### 2.3.4. Compound **4b** Weakened the Adhesion of SW620 Cells

The decisive factor of tumor metastasis is the cell adhesion to fibronectin and laminin. Weak cell adhesion is beneficial to tumor metastasis inhibition, so cell adhesion to fibronectin and laminin assay was used to evaluate the effects of different concentrations of compound **4b** and Celecoxib on the adhesion ability of SW620 cells after 24 h treatment. Results as shown in Figure 4, compound **4b** exhibited a similar ability to Celecoxib to reduce the adhesion of SW620 cells to fibronectin and laminin. 

#### 2.3.5. Xenograft Model In Vivo

In view of potent cox-2 selective inhibitory activity and anti-colon cancer proliferation activity in vitro, compound **4b** was further evaluated for anti-colon cancer activity in vivo. SW620 cells (5 × 10^6^) were subcutaneously injected into the rightwing nude mice to establish a xenograft model. When the tumor volume grows to the macroscopic size of about 100 mm^3^, 15 tumor-bearing mice were randomly divided into vehicle, Celecoxib (20 mg/kg) and compound 4b (20 mg/kg) groups. Intraperitoneal administration was performed every 2 days and tumor volume changes were recorded for 12 consecutive days. As shown in Figure 5B, tumor volume increased rapidly in the vehicle group, whereas tumor growth was significantly inhibited in two treatment groups. Among them, the tumor inhibition effect of compound **4b** (20 mg/kg) group was better than that of Celecoxib (20 mg/kg) group. After 12 days of treatment, the tumor volumes of the two treatment groups were 43.71 mm^3^ and 51.69 mm^3^, respectively. Finally, the tumors of each group were removed and weighed to calculate the ratio of tumor weight to body weight. The specific results are shown in Figure 5A,D. Compared to the vehicle group with an average ratio of tumor weight to body weight of 0.34, the other two treatment groups showed significant reduction, with compound **4b** (20 mg/kg) indicating a lighter average ratio of tumor weight to body weight (0.46). At the same time, no significant weight change was observed in the treatment group, suggesting that the compounds in these mice were nontoxic. In contrast, body weight increased slightly in the vehicle group at the later stage of treatment (Figure 5C). From the above, these results suggested that compound **4b** had potent anti-colon cancer activity in vivo. This series of representative compounds **4b**, as a selective COX-2 inhibitor for the targeted therapy of colon cancer, had prominent research prospect. 

### 2.3. Molecular Docking

In order to better study the binding mode and interaction, molecule docking of dihydropyrazole derivatives containing benzo oxygen heterocycle and sulfonamide moieties **4a**–**4z** and known COX-2 inhibitors Celecoxib about the COX-2 (PDB ID: 3LN1) enzymes were performed together. All simulations were performed around the central region of the already known COX-2 (PDB ID: 3LN1) site where Celecoxib bound. The judgment criterion is to compare the predicted combined interaction energy. The estimated value of the interaction energy of each compound were ranged from −45.96 to −33.41 kcal/mol, as displayed in the histogram of Table 5 and Figure 6. Besides, linear fitting was performed by comparing the binding energy and COX-2 inhibition. It behaved approximate linearity (y = 46.82 − 0.86x, R_2_ = 0.72) exhibited in the Figure 7. The R value is approximately equal to 0.85 and approaches to 1, which not only indicates that the fitting curve has a good credibility, but also indicates that our design idea is reasonable. Compared Celecoxib (energy value of −42.57 kcal/mol), compound **4b** was relatively superior with predicted combined interaction energy value of −45.96 kcal/mol. And their concrete combination of computer simulation was depicted in Figure 7. 

As illustrated in Figure 8A,B, Celecoxib could effectively bind at this site (XYZ axis: 30.99, −22.28 and −16.51; radius: 6.96 Å) through five hydrogen bond receptors and eleven Pi bonds, with the stronger interactional amino acids for Arg-106, Ser-339, Arg-499, Gln-178 and Leu-338. Compared with Celecoxib, compound **4b** interacts more with proteins in this region and binds better, as shown in the Figure 4C,D. Compound **4b** could effectively bind through nine hydrogen bond receptors and fourteen Pi bonds. In addition to the stronger interactional amino acids bond with Celecoxib, there are stronger interactions with His-75, Phe-504, and Val-102, which bound to the sulfa and 1,3-dioxolane domains, respectively. In general, these simulation results were consistent with the actual activity detection, and compound **4b** as a COX-2 inhibitor should be a promising effective drug. The difference in simulation between compound **4b** and Celecoxib could explain that compound **4b** was a better COX-2 inhibitor. 

## 3. Experimental Section

### 3.1. Materials and Measurements

All commercial reagents and solvents were analyzed using products produced by USA Energy Chemical company and Aladdin reagent (Shanghai, China). The melting points of all compounds were determined by an x4mp instrument. All ^1^H NMR spectra were recorded in DMSO-*d*_6_ with Bruker AM 600 MHz made by Rhenistetten Forchheim Company in Berlin, Germany as the internal standard. Published ^1^H NMR chemical shift in δ/PPM and coupling constant in Hertz. The thin layer chromatography (TLC) plate was coated with Merck silica gel 60 GF254, and the chemical reaction and purification process of the spots were observed under the light of 254/365 nm. The synthetic compounds were evaluated in vitro in the state key laboratory of pharmaceutical biotechnology of Nanjing university (Nanjing, China). 

COX-1 (human) inhibitor screening kit (catalog number 70117) andCOX-2 (human) inhibitor screening kit (catalog number 701180) were purchased from Cayman Chemical (Ann Arbor, MI, USA). 3-(4,5-Dimethylthiazol-2-yl)-2,5-diphenyltetrazolium (MTT) were purchased from Beyotime Institute of Biotechnology (Haimen, China). Annexcin V-FITC cell apoptosis assay kit (catalog No.BA11100) was purchased from BIO-BOX (Nanjing, China). Fibronectin (#F1056) and laminin (#L2020) were purchased from Sigma-Aldrich (St. Louis, MO, USA). 

### 3.2. Chemistry

#### 3.2.1. General Procedure for the Synthesis of Compound **2a**,**2b**

The protocatechuic aldehyde **1a** (1 mmol) was dissolved in anhydrous DMF (5 mL), anhydrous potassium carbonate (0.5 mmol) added. The solution was then heated to 70 °C and then dibromomethane or 1,2-dibromoethane was added dropwise. The reaction was stirred for 4 h and poured into ice water. A white precipitate (compound **2a**, **2b**) was formed which was filtered and recrystallized from ethanol [33]. 

#### 3.2.2. General Procedure for the Synthesis of Compounds **3a**–**3z**

Compound **2a**,**2b** (1 mmol) was added to 15 mL absolute ethanol solution with acetophenone derivative (1 mmol) and 40% sodium hydroxide solution (0.5 mmol). The reaction was stirred at room temperature for 10 h, and then filtered to afford compound **3a**–**3z** without further purification [34]. 

#### 3.2.3. General Procedure for the Synthesis of Compounds **4a**–**4z**

The compounds **3a**–**3z** (1 mmol), 4-sulfamethylhydrazine hydrochloride (1.2 mmol) and glacial acetic acid (0.5 mmol) were refluxed overnight in ethanol (15 mL). The reaction mixture was diluted with 50 mL water and extracted with ethyl acetate. Saturated sodium bicarbonate was used to wash the organic anhydrous layer, which was then filtered to dry by anhydrous sodium sulfate and concentrated in vacuum. The desired products **4a**–**4z** were purified by flash chromatography. 

#### 3.2.4. 4-(5-(Benzo[d][1,3]dioxol-5-yl)-3-phenyl-4,5-dihydro-1H-pyrazol-1-yl)benzenesulfonamide (**4a**)

White crystal, yield: 66.4%. m.p. 212–214 °C; ^1^H NMR (DMSO-*d*_6_, 600 MHz) δ: 7.80 (d, *J* = 7.4 Hz, 2H, ArH), 7.62 (d, *J* = 8.6 Hz, 2H, ArH), 7.48–7.40 (m, 3H, ArH), 7.11 (d, *J* = 8.6 Hz, 2H, ArH), 7.04 (s, 2H, NH_2_), 6.87 (d, *J* = 7.9 Hz, 1H, ArH), 6.78–6.75 (m, 2H, ArH), 5.90 (s, 2H, CH_2_), 5.56 (dd, *J*_1_ = 4.7, *J*_2_=4.7 Hz, 1H, 5-H), 3.91 (d, *J*_1_ = 12.4, *J*_2_ = 12.0 Hz, 1H, 4-H_b_), 3.19 (dd, *J*_1_ = 4.7, *J*_2_ = 4.6 Hz, 1H, 4-Ha). ESI-MS: *m*/*z* Calcd for C_22_H_20_N_3_O_4_S [M + H]^+^, 422.1; Found: 422.1. Anal. Calcd for C_22_H_19_N_3_O_4_S: C, 62.70; H, 4.54; N, 9.97%. Found: C, 62.92; H, 4.42; N, 9.86%. 

#### 3.2.5. 4-(5-(Benzo[d][1,3]dioxol-5-yl)-3-(p-tolyl)-4,5-dihydro-1H-pyrazol-1-yl)benzenesulfonamide (**4b**)

White crystal, yield: 76.3%. m.p. 200–201 °C; ^1^H NMR (DMSO-*d*_6_, 600 MHz) δ: 7.73 (d, *J* = 8.8 Hz, 2H, ArH), 7.59 (d, *J* = 8.7 Hz, 2H, ArH), 7.08–7.01 (m, 6H, ArH and NH_2_), 6.87 (d, *J* = 7.9 Hz, 1H, ArH), 6.77–6.73 (m, 2H, ArH), 5.98 (s, 2H, CH_2_), 5.51 (dd, *J*_1_ = 5.0, *J*_2_ = 5.1 Hz, 1H, 5-H), 3.88 (dd, *J*_1_ = 12.0, *J*_2_ = 11.9 Hz, 1H, 4-H_b_), 3.80 (s, 3H, OCH_3_), 3.15 (dd, *J*_1_ = 5.1, *J*_2_ = 5.0 Hz, 1H, 4-Ha). ESI-MS: *m*/*z* Calcd for C_23_H_22_N_3_O_4_S [M + H]^+^, 452.1; Found: 452.1. Anal. Calcd for C_23_H_21_N_3_O_4_S: C, 62.43; H, 4.86; N, 9.97%. Found: C, 62.83; H, 4.92; N, 9.82%. 

#### 3.2.6. 4-(5-(Benzo[d][1,3]dioxol-5-yl)-3-(4-ethoxyphenyl)-4,5-dihydro-1H-pyrazol-1-yl)benzenesulfonamide (**4c**)

White crystal, yield: 77.6%. m.p.182–184 °C; ^1^H NMR (DMSO-*d*_6_, 600 MHz) δ: 7.71 (d, *J* = 8.6 Hz, 2H, ArH), 7.60 (d, *J* = 8.7 Hz, 2H, ArH), 7.08–6.98 (m, 6H, ArH), 6.86 (d, *J* = 7.8Hz, 1H, ArH), 6.74 (s, 2H, NH_2_), 5.98 (s, 2H, CH_2_), 5.50 (dd, *J*_1_ = 4.9, *J*_2_ = 5.0 Hz, 1H, 5-H), 4.05 (q, *J* = 6.9 Hz, 2H, CH_2_), 3.87 (dd, *J*_1_ = 12.1, *J*_2_ = 11.8 Hz, 1H, 4-H_b_), 3.13 (dd, *J*_1_ = 5.0, *J*_2_ = 4.8 Hz, 1H, 4-Ha), 1.34 (t, *J* = 6.9 Hz, 3H, CH_3_). ESI-MS: *m*/*z* Calcd for C_24_H_24_N_3_O_5_S [M + H]^+^, 466.1; Found: 466.1. Anal. Calcd for C_24_H_23_N_3_O_5_S: C, 61.92; H, 4.98; N, 9.03%. Found: C, 61.06; H, 4.84; N, 9.32%. 

#### 3.2.7. 4-(5-(Benzo[d][1,3]dioxol-5-yl)-3-(4-iodophenyl)-4,5-dihydro-1H-pyrazol-1-yl)benzenesulfonamide (**4d**)

White crystal, yield: 58.9%. m.p. 212–214 °C; ^1^H NMR (DMSO-*d*_6_, 600 MHz) δ: 7.82 (d, *J* = 7.9 Hz, 2H, ArH), 7.62–7.54 (m, 4H, ArH), 7.16 (d, *J* = 8.5 Hz, 2H, ArH), 7.04 (s, 2H, NH_2_), 6.86 (d, *J* = 7.9 Hz, 1H, ArH), 6.54 (d, *J* = 4.2 Hz, 2H, ArH), 5.98 (s, 2H, CH_2_), 5.57 (dd, *J*_1_ = 4.7, *J*_2_ = 5.1 Hz, 1H, 5-H), 3.91 (dd, *J*_1_ = 12.2, *J*_2_ = 12.0 Hz, 1H, 4-H_b_), 3.16 (dd, *J*_1_ = 4.9, *J*_2_ = 5.0 Hz, 1H, 4-Ha). ESI-MS: *m*/*z* Calcd for C_22_H_19_IN_3_O_4_S [M + H]^+^, 548.0; Found: 548.0. Anal. Calcd for C_22_H_18_IN_3_O_4_S: C, 48.28; H, 3.31; N, 7.68%. Found: C, 48.75; H, 3.38; N, 7.52%. 

#### 3.2.8. 4-(5-(Benzo[d][1,3]dioxol-5-yl)-3-(m-tolyl)-4,5-dihydro-1H-pyrazol-1-yl)benzenesulfonamide (**4e**)

White crystal, yield: 75.8%. m.p. 187–188 °C; ^1^H NMR (DMSO-*d*_6_, 600 MHz) δ: 7.64–7.56 (m, 4H, ArH), 7.34 (t, *J* = 7.6 Hz, 1H, ArH), 7.23 (d, *J* = 7.4 Hz, 1H, ArH), 6.10 (d, *J* = 8.6 Hz, 2H, ArH), 7.04 (s, 2H, NH_2_), 6.87 (d, *J* = 7.8 Hz, 1H, ArH), 6.75 (d, *J* = 9.9 Hz, 2H, ArH), 5.98 (s, 2H, CH_2_), 5.55 (dd, *J*_1_ = 4.7, *J*_2_ = 4.8 Hz, 1H, 5-H), 3.91 (dd, *J*_1_ = 12.1, *J*_2_ = 12.1 Hz, 1H, 4-H_b_), 3.17 (dd, *J*_1_ = 4.8, *J*_2_ = 4.7 Hz, 1H, 4-Ha), 2.37 (s, 3H, CH_3_), ESI-MS: *m*/*z* Calcd for C_23_H_22_N_3_O_4_S [M + H]^+^, 436.1; Found: 436.1. Anal. Calcd for C_23_H_21_N_3_O_4_S: C, 63.43; H, 4.86; N, 9.65%. Found: C, 63.69; H, 4.91; N, 9.71%. 

#### 3.2.9. 4-(5-(Benzo[d][1,3]dioxol-5-yl)-3-(3,4-dimethylphenyl)-4,5-dihydro-1H-pyrazol-1-yl)benzenesulfonamide (**4f**)

White crystal, yield: 73.8%. m.p. 179–180 °C; ^1^H NMR (DMSO-*d*_6_, 600 MHz) δ: 7.60 (d, *J* = 8.0 Hz, 3H, ArH), 7.49 (d, *J* = 7.8 Hz, 1H, ArH), 7.21 (d, *J* = 7.8 Hz, 1H, ArH), 7.08 (d, *J* = 8.3 Hz, 2H, ArH), 7.02 (s, 2H, NH_2_), 6.86 (d, *J* = 7.8 Hz, 1H, ArH), 6.75 (s, 2H, ArH), 5.98 (s, 2H, CH_2_), 5.53 (dd, *J*_1_ = 4.4, *J*_2_ = 4.6 Hz, 1H, 5-H), 3.88 (dd, *J*_1_ = 12.1, *J*_2_ = 24.0 Hz, 1H, 4-H_b_), 3.14 (dd, *J*_1_ = 4.5, *J*_2_ = 4.4 Hz, 1H, 4-Ha). ESI-MS: *m*/*z* Calcd for C_24_H_24_N_3_O_4_S [M + H]^+^, 450.1; Found: 450.1. Anal. Calcd for C_24_H_23_N_3_O_4_S: C, 64.13; H, 5.16; N, 9.35%. Found: C, 64.47; H, 5.19; N, 9.44%. 

#### 3.2.10. 4-(5-(Benzo[d][1,3]dioxol-5-yl)-3-(3-methoxyphenyl)-4,5-dihydro-1H-pyrazol-1-yl)benzenesulfonamide (**4g**)

White crystal, yield: 76.6%. m.p. 219–221 °C; ^1^H NMR (DMSO-*d*_6_, 600 MHz) δ: 7.61 (d, *J* = 8.7 Hz, 2H, ArH), 7.37–6.99 (m, 8H, ArH), 6.87 (d, *J* = 7.9 Hz, 1H, ArH), 6.76 (s, 2H, NH_2_), 6.00 (s, 2H, CH_2_), 5.56 (dd, *J*_1_ =4.9, *J*_2_ = 5.0 Hz, 1H, 5-H), 3.91 (dd, *J*_1_ = 12.2, *J*_2_ = 12.0 Hz, 1H, 4-H_b_), 3.82 (s, 3H, CH_3_), 3.18 (dd, *J*_1_ = 5.0, *J*_2_ = 4.8 Hz, 1H, 4-Ha). ESI-MS: *m*/*z* Calcd for C_23_H_22_N_3_O_5_S [M + H]^+^, 452.1; Found: 452.1. Anal. Calcd for C_23_H_21_N_3_O_5_S: C, 61.19; H, 4.69; N, 9.31%. Found: C, 60.87; H, 4.82; N, 9.26%. 

#### 3.2.11. 4-(5-(Benzo[d][1,3]dioxol-5-yl)-3-(3-fluorophenyl)-4,5-dihydro-1H-pyrazol-1-yl)benzenesulfonamide (**4h**)

Yellow crystal, yield: 69.8%. m.p. 192–193 °C; ^1^H NMR (DMSO-*d*_6_, 600 MHz) δ: 7.95 (t, *J* = 7.6 Hz, 1H, ArH), 7.63 (d, *J* = 8.6 Hz, 2H, ArH), 7.46 (q, *J* = 7.2 Hz, 1H, ArH), 7.28 (q, *J* = 8.1 Hz, 2H, ArH), 7.11 (d, *J*=8.5 Hz, 2H, ArH), 7.06 (s, 2H, NH_2_), 6.87 (d, *J* = 7.9 Hz, 1H, ArH), 6.77 (t, *J* = 7.0 Hz, 2H, ArH), 5.99 (s, 2H, CH_2_), 5.56 (dd, *J*_1_ = 5.0, *J*_2_ = 5.0 Hz, 1H, 5-H), 4.01 (dd, *J*_1_ = 12.2, *J*_2_ = 12.2 Hz, 1H, 4-H_b_), 3.20 (dd, *J*_1_ = 4.6, *J*_2_ = 4.6 Hz, 1H, 4-Ha). ESI-MS: *m*/*z* Calcd for C_22_H_19_FN_3_O_4_S [M + H]^+^, 440.1;Found: 440.1. Anal. Calcd for C_22_H_18_FN_3_O_4_S: C, 60.13; H, 4.13; N, 9.56%. Found: C, 60.75; H, 4.79; N, 9.58%. 

#### 3.2.12. 4-(5-(Benzo[d][1,3]dioxol-5-yl)-3-(2,4-dimethylphenyl)-4,5-dihydro-1H-pyrazol-1-yl)benzenesulfonamide (**4i**)

White crystal, yield: 72.9%. m.p. 180–182 °C; ^1^H NMR (DMSO-*d*_6_, 600 MHz) δ: 7.60 (d, *J* = 8.6 Hz, 2H, ArH), 7.35 (d, *J* = 7.9 Hz, 1H, ArH), 7.17 (s, 1H, ArH), 7.10–7.03 (m, 5H, ArH and NH_2_), 6.87 (d, *J* = 7.7 Hz, 1H, ArH), 6.77–6.74 (m, 2H, ArH), 6.98 (s, 2H, CH_2_), 5.48 (dd, *J*_1_ = 4.8, *J*_2_ = 4.8 Hz, 1H, 5-H), 3.98 (dd, *J*_1_ = 12.0, *J*_2_=11.8 Hz, 1H, 4-H_b_), 3.20 (dd, *J*_1_ = 4.9, *J*_2_ = 4.8 Hz, 1H, 4-Ha), 2.67 (s, 3H, CH_3_), 2.31 (s, 3H, CH_3_). ESI-MS: *m*/*z* Calcd for C_24_H_24_N_3_O_4_S [M + H]^+^, 450.1; Found: 450.1. Anal. Calcd for C_24_H_23_N_3_O_4_S: C, 64.13; H, 5.16; N, 9.35%. Found: C, 64.72; H, 5.19; N, 9.51%. 

#### 3.2.13. 4-(5-(Benzo[d][1,3]dioxol-5-yl)-3-(2-fluorophenyl)-4,5-dihydro-1H-pyrazol-1-yl)benzenesulfonamide (**4j**)

White crystal, yield: 66.5%. m.p. 180–181 °C; ^1^H NMR (DMSO-*d*_6_, 600 MHz) δ: 7.95 (t, *J* = 7.6 Hz, 1H, ArH), 7.63 (d, *J* = 8.6 Hz, 2H, ArH), 7.46 (q, *J* = 7.2 Hz, 1H, ArH), 7.28 (q, *J* = 8.1 Hz, 2H, ArH), 7.11 (d, *J* = 8.5 Hz, 2H, ArH), 7.06 (s, 2H, NH_2_), 6.87 (d, *J* = 7.9 Hz, 1H, ArH), 6.77 (t, *J* = 7.0 Hz, 2H, ArH), 5.99 (s, 2H, CH_2_), 5.56 (dd, *J*_1_ = 5.0, *J*_2_ = 5.0 Hz, 1H, 5-H), 4.01 (dd, *J*_1_ = 12.2, *J*_2_ = 12.2 Hz, 1H, 4-H_b_), 3.20 (dd, *J*_1_ = 4.6, *J*_2_ = 4.6 Hz, 1H, 4-Ha). ESI-MS: *m*/*z* Calcd for C_22_H_19_FN_3_O_4_S [M + H]^+^, 440.1; Found: 440.1. Anal. Calcd for C_22_H_18_FN_3_O_4_S: C, 60.13; H, 4.13; N, 9.56%. Found: C, 60.61; H, 4.59; N, 9.53%. 

#### 3.2.14. 4-(5-(2,3-Dihydrobenzo[b][1,4]dioxin-6-yl)-3-phenyl-4,5-dihydro-1H-pyrazol-1-yl)benzenesulfonamide (**4k**)

White crystal, yield: 73.5%. m.p. 156–157 °C; ^1^H NMR (DMSO-*d*_6_, 600 MHz) δ: 7.79 (d, *J* = 6.8 Hz, 2H, ArH), 7.61 (d, *J* = 8.9 Hz, 2H, ArH), 7.47–7.40 (m, 3H, ArH), 7.09 (t, *J* = 8.9 Hz, 2H, ArH), 6.99 (s, 2H, NH_2_), 6.82 (d, *J* = 8.4 Hz, 1H, ArH), 6.72–6.99 (m, 2H, ArH),5.54 (dd, *J*_1_ = 4.8, *J*_2_ = 4.8 Hz, 1H, 5-H), 4.18 (s, 4H, CH_2_), 3.91 (dd, *J*_1_ = 12.1, *J*_2_ = 12.0 Hz, 1H, 4-H_b_), 3.17 (dd, *J*_1_ = 4.9, *J*_2_ = 3.6 Hz, 1H, 4-Ha). ESI-MS: *m*/*z* Calcd for C_23_H_22_N_3_O_4_S [M + H]^+^, 435.1; Found: 435.1. Anal. Calcd for C_23_H_21_N_3_O_4_S: C, 63.43; H, 4.86; N, 9.65%. Found: C, 63.52; H, 4.97; N, 9.71%. 

#### 3.2.15. 4-(5-(2,3-Dihydrobenzo[b][1,4]dioxin-6-yl)-3-(p-tolyl)-4,5-dihydro-1H-pyrazol-1-yl)benzenesulfonamide (**4l**)

White crystal, yield: 75.6%. m.p. 153–154 °C; ^1^H NMR (DMSO-*d*_6_, 600 MHz) δ: 7.68 (d, *J* = 8.1 Hz, 2H, ArH), 7.59 (d, *J* = 8.9 Hz, 2H, ArH), 7.26 (d, *J* = 8.0 Hz, 2H, ArH), 7.06 (t, *J* = 8.9 Hz, 1H, ArH), 6.99 (s, 2H, NH_2_), 6.82 (d, *J* = 8.6 Hz, 1H, ArH), 6.70-6.68 (m, 2H, ArH), 5.51 (dd, *J*_1_ = 4.7, *J*_2_ = 4.7 Hz, 1H, 5-H), 4.18 (s, 4H, CH_2_), 3.88 (dd, *J*_1_ = 12.0, *J*_2_ = 11.8 Hz, 1H, 4-H_b_), 3.13 (dd, *J*_1_ = 4.9, *J*_2_ = 4.8 Hz, 1H, 4-Ha), 2.34 (s, 3H, CH_3_). ESI-MS: *m*/*z* Calcd for C_24_H_24_N_3_O_4_S [M + H]^+^, 450.1; Found:450.1. Anal. Calcd for C_24_H_23_N_3_O_4_S: C, 64.13; H, 5.16; N, 9.35%. Found: C, 64.59; H, 5.27; N, 9.48%. 

#### 3.2.16. 4-(5-(2,3-Dihydrobenzo[b][1,4]dioxin-6-yl)-3-(4-methoxyphenyl)-4,5-dihydro-1H-pyrazol-1-yl)benzenesulfonamide (**4m**)

White crystal, yield: 75.6%. m.p. 207–208 °C; ^1^H NMR (DMSO-*d*_6_, 600 MHz) δ: 7.73 (d, *J* = 8.8 Hz, 2H, ArH), 7.59 (d, *J* = 8.7 Hz, 2H, ArH), 7.08–7.01 (m, 6H, ArH and NH_2_), 6.87 (d, *J* = 7.9 Hz, 1H, ArH), 6.77–6.73 (m, 2H, ArH), 5.49 (dd, *J*_1_ = 5.0, *J*_2_ = 5.1 Hz, 1H, 5-H), 4.19 (s, 4H, CH_2_), 3.87 (dd, *J*_1_ = 12.0, *J*_2_ = 11.9 Hz, 1H, 4-H_b_), 3.81(s, 3H, OCH_3_), 3.13 (dd, *J*_1_ = 5.1, *J*_2_ = 5.0 Hz, 1H, 4-Ha). ESI-MS: *m*/*z* Calcd for C_24_H_24_N_3_O_5_S [M + H]^+^, 466.1; Found: 466.1. Anal. Calcd for C_24_H_23_N_3_O_5_S: C, 64.13; H, 5.16; N, 9.35%. Found: C, 64.59; H, 5.27; N, 9.48%. 

#### 3.2.17. 4-(5-(2,3-Dihydrobenzo[b][1,4]dioxin-6-yl)-3-(4-ethoxyphenyl)-4,5-dihydro-1H-pyrazol-1-yl)benzenesulfonamide (**4n**)

White crystal, yield: 59.8%. m.p. 193–194 °C; ^1^H NMR (DMSO-*d*_6_, 600 MHz) δ: 7.71 (d, *J* = 8.6 Hz, 2H, ArH), 7.60 (d, *J* = 8.7 Hz, 2H, ArH), 7.07–6.98 (m, 6H, ArH), 6.81 (d, *J* = 8.0Hz, 1H, ArH), 6.70 (*s*, 2H, NH_2_), 5.48 (dd, *J*_1_ = 5.0, *J*_2_ = 4.7 Hz, 1H, 5-H), 4.18 (s, 4H, CH_2_), 4.05 (q, *J* = 6.8 Hz, 2H, CH_2_), 3.86 (dd, *J*_1_ = 12.0, *J*_2_ = 11.7 Hz, 1H, 4-H_b_), 3.12 (dd, *J*_1_ = 4.7, *J*_2_ = 4.6 Hz, 1H, 4-Ha), 1.34 (t, *J* = 6.9 Hz, 3H, CH_3_). ESI-MS: *m*/*z* Calcd for C_25_H_26_N_3_O_5_S [M + H]^+^, 480.1. Anal. Calcd for C_25_H_25_N_3_O_5_S: C, 62.62; H, 5.25; N, 8.76%. Found: C, 63.53; H, 5.49; N, 8.98%. 

#### 3.2.18. 4-(5-(2,3-Dihydrobenzo[b][1,4]dioxin-6-yl)-3-(4-fluorophenyl)-4,5-dihydro-1H-pyrazol-1-yl)benzenesulfonamide (**4o**)

Yellow crystal, yield: 67.8%. m.p. 195–196 °C; ^1^H NMR (DMSO-*d*_6_, 600 MHz) δ: 7.84 (q, *J* = 5.6 Hz, 2H, ArH), 7.60 (d, *J* = 8.9 Hz, 2H, ArH), 7.29 (t, *J* = 8.8 Hz, 2H, ArH), 7.08 (t, *J* = 8.9 Hz, 2H, ArH), 6.99 (s, 2H, NH_2_), 6.82 (q, 2, *J* = 8.1 Hz, 2H, ArH), 6.70 (t, *J* = 8.0 Hz, 2H, ArH), 5.54 (dd, *J*_1_ = 4.8, *J*_2_ = 4.8 Hz, 1H, 5-H), 4.19 (s, 4H, CH_2_), 3.91 (dd, *J*_1_ = 12.1, *J*_2_ = 11.9 Hz, 1H, 4-H_b_), 3.16 (dd, *J*_1_ = 5.0, *J*_2_ = 4.8 Hz, 1H, 4-Ha). ESI-MS: *m*/*z* Calcd for C_23_H_21_FN_3_O_4_S [M + H]^+^, 454.1; Found: 454.1. Anal. Calcd for C_23_H_20_FN_3_O_4_S: C, 60.92; H, 4.45; N, 9.27%. Found: C, 61.63; H, 4.58; N, 9.13%. 

#### 3.2.19. 4-(3-(4-Chlorophenyl)-5-(2,3-dihydrobenzo[b][1,4]dioxin-6-yl)-4,5-dihydro-1H-pyrazol-1-yl)benzenesulfonamide (**4p**)

Yellow crystal, yield: 73.9%. m.p. 146–148 °C; ^1^H NMR (DMSO-*d*_6_, 600 MHz) δ: 7.79 (d, *J* = 8.6 Hz, 2H, ArH), 7.61 (d, *J* = 8.8 Hz, 2H, ArH), 7.50 (d, *J* = 8.6 Hz, 2H, ArH), 7.11–7.08 (m, 3H, ArH), 6.82 (d, *J* = 8.0 Hz, 2H, ArH), 6.99 (s, 2H, NH_2_), 5.55 (dd, *J*_1_ = 4.8, *J*_2_=4.8 Hz, 1H, 5-H), 4.19 (s, 4H, CH_2_), 3.89 (dd, *J*_1_ = 7.0, *J*_2_ = 12.0 Hz, 1H, 4-H_b_), 3.16 (dd, *J*_1_ = 5.0, *J*_2_ = 4.9 Hz, 1H, 4-Ha). ESI-MS: *m*/*z* Calcd for C_23_H_21_ClN_3_O_4_S [M + H]^+^, 470.1; Found: 470.1. Anal. Calcd for C_23_H_20_ClN_3_O_4_S: C, 58.78; H, 4.29; N, 8.94%. Found: C, 59.97; H, 4.48; N, 9.18%. 

#### 3.2.20. 4-(3-(4-Bromophenyl)-5-(2,3-dihydrobenzo[b][1,4]dioxin-6-yl)-4,5-dihydro-1H-pyrazol-1-yl)benzenesulfonamide (**4q**)

White crystal, yield: 63.5%. m.p. 153–154 °C; ^1^H NMR (DMSO-*d*_6_, 600 MHz) δ: 7.72 (d, *J* = 8.6 Hz, 2H, ArH), 7.61 (q, *J* = 8.8 Hz, 2H, ArH), 7.09 (t, *J* = 8.8 Hz, 2H, ArH), 6.99 (s, 2H, NH_2_), 6.81 (d, *J* = 8.0 Hz, 2H, ArH), 5.55 (dd, *J*_1_ = 4.9, *J*_2_ = 4.9 Hz, 1H, 5-H), 4.19 (s, 4H, CH_2_), 3.91(dd, *J*_1_ = 12.2, *J*_2_ = 12.0 Hz, 1H, 4-H_b_), 3.16 (dd, *J*_1_ = 5.0, *J*_2_ = 4.9 Hz, 1H, 4-Ha). ESI-MS: *m*/*z* Calcd for C_23_H_21_BrN_3_O_4_S [M + H]^+^, Found: 514.0. Anal. Calcd for C_23_H_20_BrN_3_O_4_S: C, 53.70; H, 3.92; N, 8.17%. Found: C, 53.54; H, 3.18; N, 8.43%. 

#### 3.2.21. 4-(5-(2,3-Dihydrobenzo[b][1,4]dioxin-6-yl)-3-(4-iodophenyl)-4,5-dihydro-1H-pyrazol-1-yl)benzenesulfonamide (**4r**)

Black crystal, yield: 72.3%. m.p. 187–189 °C; ^1^H NMR (DMSO-*d*_6_, 600 MHz) δ: 7.79 (d, *J* = 8.1 Hz, 2H, ArH), 7.63 (d, *J* = 7.6 Hz, 2H, ArH), 7.55 (d, *J* = 8.0 Hz, 2H, ArH), 7.11 (d, *J* = 8.6 Hz, 2H, ArH), 6.82 (d, *J* = 8.1 Hz, 2H, ArH), 6.70 (s, 2H, NH_2_), 5.53 (dd, *J*_1_ = 4.6, *J*_2_ = 4.7 Hz, 1H, 5-H), 4.18 (s, 4H, CH_2_), 3.87(dd, *J*_1_ = 12.4, *J*_2_ = 12.0 Hz, 1H, 4-H_b_), 3.13 (dd, *J*_1_ = 4.7, *J*_2_ = 4.5 Hz, 1H, 4-Ha). ESI-MS: *m*/*z* Calcd for C_23_H_21_IN_3_O_4_S [M + H]^+^, 562.0; Found: 562.0. Anal. Calcd for C_23_H_20_IN_3_O_4_S: C, 49.21; H, 3.59; N, 7.49%. Found: C, 49.88; H, 3.95; N, 7.36%. 

#### 3.2.22. 4-(5-(2,3-Dihydrobenzo[b][1,4]dioxin-6-yl)-3-(3,4-dimethylphenyl)-4,5-dihydro-1H-pyrazol-1-yl)benzenesulfonamide (**4s**)

White crystal, yield: 72.3%. m.p. 166–168 °C; ^1^H NMR (DMSO-*d*_6_, 600 MHz) δ: 7.60 (d, *J* = 8.7 Hz, 3H, ArH), 7.48 (d, *J* = 7.8 Hz, 2H, ArH), 7.20 (d, *J* = 7.9 Hz, 1H, ArH), 7.07 (t, *J* = 8.7 Hz, 3H, ArH), 6.81 (d, *J* = 8.4 Hz, 1H, ArH), 6.90 (s, 2H, NH_2_), 5.50 (dd, *J*_1_ = 4.5, *J*_2_=4.6 Hz, 1H, 5-H), 4.19 (s, 4H, CH_2_), 3.86 (dd, *J*_1_ = 12.0, *J*_2_ = 11.8 Hz, 1H, 4-H_b_), 3.12 (dd, *J*_1_ = 4.6, *J*_2_ = 4.5 Hz, 1H, 4-Ha), 2.27 (s, 3H, CH_3_), 2.26 (s, 3H, CH_3_). ESI-MS: *m*/*z* Calcd for C_25_H_26_N_3_O_4_S [M + H]^+^, 464.1; Found: 464.1. Anal. Calcd for C_25_H_25_N_3_O_4_S: C, 64.78; H, 5.44; N, 9.07%. Found: C, 65.92 H, 5.69; N, 9.34%. 

#### 3.2.23. 4-(5-(2,3-Dihydrobenzo[b][1,4]dioxin-6-yl)-3-(4-fluoro-3-methylphenyl)-4,5-dihydro-1H-pyrazol-1-yl)benzenesulfonamide (**4t**)

Yellow crystal, yield: 71.4%. m.p. 159–161 °C; ^1^H NMR (DMSO-*d*_6_, 600 MHz) δ: 7.78 (q, *J* = 8.9 Hz, 1H, ArH), 7.64–7.60 (m, 3H, ArH), 7.48–7.40 (m, 1H, ArH), 7.11(t, *J* = 8.9 Hz, 2H, ArH), 7.09 (s, 2H, NH_2_), 6.82 (t, *J* = 8.2 Hz, 2H, ArH), 6.72–6.69 (m, 2H, ArH), 5.54 (dd, *J*_1_ = 4.9, *J*_2_ = 5.4 Hz, 1H, 5-H), 4.19 (s, 4H, CH_2_), 3.86 (dd, *J*_1_ = 12.1, *J*_2_ = 11.9 Hz, 1H, 4-H_b_), 3.15 (dd, *J*_1_ = 5.0, *J*_2_ = 4.8 Hz, 1H, 4-Ha), 2.35 (s, 3H, CH_3_), ESI-MS: *m*/*z* Calcd for C_24_H_23_FN_3_O_4_S [M + H]^+^, 468.1; Found: 468.1. Anal. Calcd for C_24_H_22_FN_3_O_4_S: C, 61.66; H, 4.74; N, 8.99%. Found: C, 62.42; H, 4.58; N, 9.29%. 

#### 3.2.24. 4-(5-(2,3-Dihydrobenzo[b][1,4]dioxin-6-yl)-3-(3-methoxyphenyl)-4,5-dihydro-1H-pyrazol-1-yl)benzenesulfonamide (**4u**)

White crystal, yield: 70.1%. m.p. 146–147 °C; ^1^H NMR (DMSO-*d*_6_, 600 MHz) δ: 7.60 (d, *J* = 8.9 Hz, 2H, ArH), 7.34 (d, *J* = 4.9 Hz, 2H, ArH), 7.11–7.00 (m, 3H, ArH), 6.99 (s, 2H, NH_2_), 6.82 (d, *J* = 8.4 Hz, 2H, ArH), 6.71–6.69 (m, 2H, ArH), 5.54 (dd, *J*_1_ = 4.7, *J*_2_ = 4.9 Hz, 1H, 5-H), 4.19 (s, 4H, CH_2_), 3.89 (dd, *J*_1_ = 12.1, *J*_2_ = 12.0 Hz, 1H, 4-H_b_), 3.80 (s, 3H, OCH_3_), 3.17 (dd, *J*_1_ = 4.9, *J*_2_ = 4.8 Hz, 1H, 4-Ha). ESI-MS: *m*/*z* Calcd for C_24_H_24_N_3_O_5_S [M + H]^+^, Found: 466.1. Anal. Calcd for C_24_H_23_N_3_O_5_S: C, 61.92; H, 4.98; N, 9.03%. Found: C, 62.37; H, 5.16; N, 9.31%. 

#### 3.2.25. 4-(5-(2,3-Dihydrobenzo[b][1,4]dioxin-6-yl)-3-(3-fluorophenyl)-4,5-dihydro-1H-pyrazol-1-yl)benzenesulfonamide (**4v**)

Yellow crystal, yield: 68.7%. m.p. 200–201 °C; ^1^H NMR (DMSO-*d*_6_, 600 MHz) δ: 7.84 (q, *J* = 5.6 Hz, 2H, ArH), 7.60 (d, *J* = 8.9 Hz, 2H, ArH), 7.29 (t, *J* = 8.8 Hz, 2H, ArH), 7.08 (t, *J* = 8.9 Hz, 2H, ArH), 6.99 (s, 2H, NH_2_), 6.82 (q, 2, *J*=8.1 Hz, 2H, ArH), 6.70 (t, *J* = 8.0 Hz, 2H, ArH), 5.54 (dd, *J*_1_ = 4.8, *J*_2_ = 4.8 Hz, 1H, 5-H), 4.19 (s, 4H, CH_2_), 3.91 (dd, *J*_1_ = 12.1, *J*_2_ = 11.9 Hz, 1H, 4-H_b_), 3.16 (dd, *J*_1_ = 5.0, *J*_2_ = 4.8 Hz, 1H, 4-Ha). ESI-MS: *m*/*z* Calcd for C_22_H_21_FN_3_O_4_S [M + H]^+^, 454.1; Found: 454.1. Anal. Calcd for C_22_H_20_FN_3_O_4_S: C, 60.92; H, 4.45; N, 9.27%. Found: C, 62.21; H, 4.69; N, 9.44%. 

#### 3.2.26. 4-(3-(3,5-Difluorophenyl)-5-(2,3-dihydrobenzo[b][1,4]dioxin-6-yl)-4,5-dihydro-1H-pyrazol-1-yl)benzenesulfonamide (**4w**)

White crystal, yield: 63.4%. m.p. 182–183 °C; ^1^H NMR (DMSO-*d*_6_, 600 MHz) δ: 7.62 (q, *J* = 8.9 Hz, 2H, ArH), 7.50–7.47 (m, 2H, ArH), 7.31–7.26 (m, 1H, ArH), 7.15 (t, *J* = 8.9 Hz, 2H, ArH), 7.10 (s, 2H, NH_2_), 6.82 (t, *J* = 8.2 Hz, 2H, ArH), 6.72–6.69 (m, 2H, ArH), 5.60 (dd, *J*_1_ = 4.9, *J*_2_ = 5.4 Hz, 1H, 5-H), 4.19 (s, 4H, CH_2_), 3.88 (dd, *J*_1_ = 12.1, *J*_2_ = 11.9 Hz, 1H, 4-H_b_), 3.19 (dd, *J*_1_ = 5.0, *J*_2_ = 4.8 Hz, 1H, 4-Ha). ESI-MS: *m*/*z* Calcd for C_23_H_20_F_2_N_3_O_4_S [M + H]^+^, 472.1; Found: 472.1. Anal. Calcd for C_23_H_19_F_2_N_3_O_4_S: C, 58.59; H, 4.06; N, 8.91%. Found: C, 58.96; H, 4.74; N, 8.88%. 

#### 3.2.27. 4-(3-(2,3-Dichlorophenyl)-5-(2,3-dihydrobenzo[b][1,4]dioxin-6-yl)-4,5-dihydro-1H-pyrazol-1-yl)benzenesulfonamide (**4x**)

Yellow crystal, yield: 73.5%. m.p. 171–173 °C; ^1^H NMR (DMSO-*d*_6_, 600 MHz) δ: 7.97 (d, *J* = 1.9 Hz, 2H, ArH), 7.76–7.61 (m, 5H, ArH), 7.24–7.06 (m, 4H, ArH), 6.82–6.68 (m, 3H, ArH and NH_2_), 5.59 (dd, *J*_1_ = 4.8, *J*_2_ = 5.0 Hz, 1H, 5-H), 4.19 (s, 4H, CH_2_), 3.89 (dd, *J*_1_ = 12.0, *J*_2_ = 24.0 Hz, 1H, 4-H_b_), 3.19 (dd, *J*_1_ = 5.0, *J*_2_ = 4.9 Hz, 1H, 4-Ha). ESI-MS: *m*/*z* Calcd for C_23_H_20_Cl_2_N_3_O_4_S [M + H]^+^, 504.1; Found: 504.1. Anal. Calcd for C_23_H_19_Cl_2_N_3_O_4_S: C, 54.77; H, 3.80; N, 8.33%. Found: C, 54.96; H, 3.95; N, 8.53%. 

#### 3.2.28. 4-(5-(2,3-Dihydrobenzo[b][1,4]dioxin-6-yl)-3-(2-fluorophenyl)-4,5-dihydro-1H-pyrazol-1-yl)benzenesulfonamide (**4y**)

Yellow crystal, yield: 69.4%. m.p. 191–192 °C; ^1^H NMR (DMSO-*d*_6_, 600 MHz) δ: 7.94 (t, *J* = 7.6 Hz, 1H, ArH), 7.63 (d, *J* = 8.6 Hz, 2H, ArH), 7.45 (q, *J* = 7.1 Hz, 1H, ArH), 7.27 (q, *J* = 8.7 Hz, 2H, ArH), 7.11 (d, *J* = 9.0 Hz, 4H, ArH), 7.82 (d, *J* = 8.1 Hz, 4H, ArH), 6.72 (s, 2H, NH_2_), 5.53 (dd, *J*_1_ = 4.7, *J*_2_ = 4.8 Hz, 1H, 5-H), 4.19 (s, 4H, CH_2_), 3.99 (dd, *J*_1_ = 12.1, *J*_2_ = 12.1 Hz, 1H, 4-H_b_), 3.17 (dd, *J*_1_=2.5, *J*_2_ = 2.2 Hz, 1H, 4-Ha). ESI-MS: *m*/*z* Calcd for C_23_H_21_FN_3_O_4_S [M + H]^+^, 454.1; Found: 454.1. Anal. Calcd for C_23_H_20_FN_3_O_4_S: C, 60.92; H, 4.45; N, 9.27%. Found: C, 61.39; H, 4.69; N, 9.31%. 

#### 3.2.29. 4-(3-(2,4-Difluorophenyl)-5-(2,3-dihydrobenzo[b][1,4]dioxin-6-yl)-4,5-dihydro-1H-pyrazol-1-yl)benzenesulfonamide (**4z**)

White crystal, yield: 65.3%. m.p. 168–169 °C; ^1^H NMR (DMSO-*d*_6_, 600 MHz) δ: 7.97 (q, *J* = 7.6 Hz, 1H, ArH), 7.61 (d, *J* = 7.9 Hz, 2H, ArH), 7.36 (q, *J* = 9.7 Hz, 1H, ArH), 7.20 (t, *J* = 8.4 Hz, 2H, ArH), 7.10 (d, *J* = 9.0 Hz, 2H, ArH), 7.08 (s, 2H, NH_2_), 6.81(d, *J* = 8.4 Hz, 1H, ArH), 6.70 (d, *J* = 10.0 Hz, 2H, ArH), 5.53 (dd, *J*_1_ = 4.5, *J*_2_ = 4.7 Hz, 1H, 5-H), 4.19 (s, 4H, CH_2_), 3.98 (dd, *J*_1_ = 12.3, *J*_2_ = 12.2Hz, 1H, 5-H), 3.44 (dd, *J*_1_ = 6.8, *J*_2_ = 7.4 Hz, 1H, 4-H_b_), 3.14 (dd, *J*_1_=2.5, *J*_2_ = 4.3Hz, 1H, 4-Ha). ESI-MS: *m*/*z* Calcd for C_23_H_20_F_2_N_3_O_4_S [M + H]^+^, 472.1; 472.1. Anal. Calcd for C_23_H_19_F_2_N_3_O_4_S: C, 58.59; H, 4.06; N, 8.91%. Found: C, 59.67; H, 4.52; N, 9.06%. 

### 3.3. Biological Assays

#### 3.3.1. COX-1/COX-2 Inhibition Assay

The ability of the synthetic compounds to inhibit COX-1 and COX-2 was evaluated by COX-1/COX-2 ELISA kit (no. 701170/701180, Cayman Chemistry). COX-1 or COX-2 enzyme was pre-incubated with test compounds at 0 μM, 1 μM, 10 μM and 100 μM in the supplied reaction buffer (0.1 M Tris–HCl, pH 8.0, 5 mM EDTA, 2 mM phenol and 1 μM heme) for 10 min at 37 °C. The reactions were initiated by adding arachidonic acid to a final concentration of 100 μM and incubated 2 min at 37 °C for. Then 1 M HCl was added to the reaction mixture to stop the reaction, and one-tenth of the volume of saturated stannous chloride (50 mg/mL) was added. The reaction mixture was incubated at room temperature for 5 min, and PGF2 and PGH2 were produced. 

#### 3.3.2. Cell Culture

MCF-7 (human breast cell line), HeLa (human cervical cell line), A549 (human lung cell line), HepG2 (human liver cell line), SW620 (human colorectal cell line) and a non-cancer cell line, NCM460 (Human colonic epithelial cells) were preserved at the State Key Laboratory of Pharmaceutical Biotechnology, Nanjing University (Nanjing, China). Cells were maintained in Dulbecco′s modified Eagle′s medium (DMEM) (HyClone, Logan, UT, USA) with 10% fetal bovine serum (FBS, BI) (HyClone, Logan, UT, USA), 100 U/mL penicillin and 100 mg/mL streptomycin (Hyclone, Logan, UT, USA), and incubated at 37 °C in a humidified atmosphere containing 5% CO_2_. 

#### 3.3.3. Cell Proliferation Assay

The antiproliferative activity of the synthetic compounds against the SW620, MCF-7, HeLa, A549, HepG2, and NUM460 cell lines was evaluated by a colorimetric assay based on a modified standard (MTT). Test cell lines were seeded at a density of 1 × 10^4^/well in 96-well plates and incubated for 12 h at 37 °C in DMEM supplemented with 10% fetal bovine serum. All test compounds dissolved in DMSO were then treated to cells at 0 μM, 0.1 μM, 1 μM, 10 μM, and 100 μM and incubated for 48 h at 37 °C in 5% CO_2_. Thereafter, MTT (5 mg/mL in PBS) was added to each well and incubated for 4 h. 150 μL of DMSO was added to each well and shaken with a shaker. The absorbance (OD 570 nm) was read at 630 nm on an ELISA reader (ELx800, BioTek, montpelier, VT, USA). IC_50_ values for the compounds were calculated by comparison to DMSO treated control wells. Three replicate wells were used for each drug concentration. Each assay was performed three times. 

#### 3.3.4. Cell Apoptosis Assay

SW620 cells were seeded in the 6-well plate (2 × 10^5^ cells/m^2^) and treated with compound **4b** to induce apoptosis at final concentrations of 0, 2, 4, and 8 μM for 24 h. After incubation, the cells were extracted with trypsin and washed with cold PBS twice. Cells are then bound to 500 μL buffer and then annexin-v/FITC (5 μL) and PI (5 μL) were added to the medium for another 10 min to incubate darkness. Staining cells were analyzed by flow cytometry (Becton Dickinson, Franklin, NJ, USA). Flowjo 7.6.1 software (Emerald Biotech Co. Ltd., Hangzhou, China) was used for statistical analysis. 

#### 3.3.5. Cell Adhesion Assay

In the cell adhesion experiment, the 96-well plate was coated with 50 L fibrin and laminin (10 g/mL) (Sigma-Aldrich, St. Louis, MI, USA) at a constant temperature of 4 °C for 12 h, closed with 0.2% BSA for 2 h, and washed 3 times at room temperature. Afterwards, A549 cells treated with 4 days and celecoxib for 24 h each were plated to the coated wells (10^4^ per well) and incubated at 37 °C, 5% CO_2_ for 40 min. SW620 cells were allowed to adhere to the coated surface, washed intensively with PBS three times to remove non-adherent cells, and then incubated in 5 g/mL MTT in complete medium at 37 °C for 4 h. Then, mtt-treated cells were treated in DMSO, and the absorbance was measured by ELISA reader. 

#### 3.3.6. Xenograft Model In Vivo

Nude mice aged 6-8 weeks were purchased from the model animal research center of nanjing university and bred in a specific non-pathogenic environment. A xenograft model was established by subcutaneous injection of SW620 cells (5 × 10^8^)100 L DMEM into the right wing (18–22 g) of nude mice. When the tumor mass was visible and the tumor size was close to 100 mm^3^.Tumor-bearing nude mice were randomly divided into 3 groups (5 mice/group), vehicle-mounted treatment group, compound **4b** (20 mg/kg) treatment group and Celecoxib (20 mg/kg) treatment group. Vehicle treatment groups were treated with polyethylene glycol (containing 1% DMSO). The administration method was intraperitoneal injection every 2 days for consecutive 12 days. After administration, weighed every two days and measured tumor volume with a vernier caliper. Animal welfare and experimental procedures were carried out in accordance with Laboratory animal-Guideline for ethical review of animal welfare (China national standardization management committee, GB/T 35892-2018) and Laboratory animals-General requirements for animal experiment (China national standardization management committee, GB/T 35823-2018). All the animal experiments were approved by Nanjing University Animal Care and Use Committee (NJU-ACUC) and made to minimize suffering and to reduce the number of animals used. 

### 3.4. Docking Simulations

The crystal structure of the protein complex was retrieved from the RCSB protein database (PDB code: 3LN1). Then, the three-dimensional structure of the compound was constructed using Chem 3D Ultra 14.0 software (Cambridge Software, Boston, MA, USA), and then energy minimization was performed using MMFF94, iterating 5000 times with a minimum RMS gradient of 0.10. Compound **4b** and Celecoxib molecules were docked into a three-dimensional complex structure of COX-2 using the Discovery Studio (Discovery Studio 4.5, Accelrys, Co. Ltd., Beijing, China) software via the graphical user interface DS-CDOCKER protocol. All bound water and ligand were removed from the protein and polar hydrogen was added to the protein. Briefly, we defined the COX-2 complex as a receptor and then replaced the Celecoxib molecule with compound **4b**, mimicking the entry of the compound into the site where COX-2 binds to Celecoxib. 

## 4. Conclusions

Overexpression of COX-2 was clearly associated with the development of various types of cancer, and the concept of COX-2 as a possible target was a promising therapeutic strategy. Therefore, a series of novel dihydropyrazole derivatives containing benzo oxygen heterocycle and sulfonamide moieties **4a**–**4z** were designed to be COX-2 inhibitors. Most of the synthetic compounds exhibited strong selective and inhibitory activity, and compound **4b** exhibited the most potent inhibitory activity against COX-2 and antiproliferative activity on SW620 cells with IC_50_ values of 0.86 μM. In further research, representative compound **4b** induced apoptosis in SW620 cells in a dose-dependent manner and attenuated its adhesion. Finally, the in vivo anti-colon cancer activity of compound **4b** was verified in SW620 xenograft mouse models. In conclusion, it was hoped that this study would be beneficial to the research of COX-2 inhibitors in the treatment for colorectal carcinoma.

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
