# Peer review of "Dihydropyrazole Derivatives Containing Benzo Oxygen Heterocycle and Sulfonamide Moieties Selectively and Potently Inhibit COX-2: Design, Synthesis, and Anti-Colon Cancer Activity Evaluation"

_molecules, 2019, doi:10.3390/molecules24091685_

Round 1
Reviewer 1 Report
The manuscript by Xiao-Qiang Yan et al. reports the synthesis of a wide range of dihydropyrazole derivatives containing benzodioxole rings. The compounds were well characterized by conventional techniques and their potent COX inhibition activity was demonstrated. The manuscript presents interest for readers specializing in medicinal chemistry and fits the scope of Molecules journal.
The remarks on material presentation are the following:
1. In the introduction part, some context of the previous works of the authors should be given. For example, I found at least one paper (Bioorg and Med Chem Lett, 2013, 23, 1091) from the same group of authors dealing with enzyme inhibition by structurally similar dihydropyrazole derivatives. Possibly, there are other published papers and a short overview of them would give the reader a better outlook of the work.
2. Compounds 2a-b were reported in the literature before, so the authors should give references for the preparation procedures that they used. Same goes for compounds 3a-z, it is unclear from the paper if some of these compound were prepared for the first time. If that is the case, the authors should give full characterization for the new compounds. If all of these compounds were reported before, appropriate references should be given.
3. CheckCIF report for structure 4b contains a lot of level A and level B alerts, which should be dealt with (or explanations for reasons why a specific alert could not be eliminated should be given) before this structure can be published.
4. Furthermore, the overall quality of structure solution is not very good; some displacement ellipsoids are rather distorted, indicating that the disorder model used for structure solution can be improved.
5. The resolution of images for NMR spectra given in the supplementary material is to low, which makes them almost useless. I suggest preparing a PDF document with vector images of NMR spectra.
Author Response
Response to Reviewers
Dear Reviewers,
Many thanks for your comments. I really appreciate your kindly recommendations with our manuscript. According to the comments, we realize that our manuscript should be improved by a major revision. Therefore, we have revised our manuscript according to your comments. We hope the revised manuscript would be better. The main points of response to reviewers are as follow:
Reviewers' comments:
Reviewer #1: The manuscript by Xiao-Qiang Yan et al. reports the synthesis of a wide range of dihydropyrazole derivatives containing benzodioxole rings. The compounds were well characterized by conventional techniques and their potent COX inhibition activity was demonstrated. The manuscript presents interest for readers specializing in medicinal chemistry and fits the scope of Molecules journal.
The remarks on material presentation are the following:
1. In the introduction part, some context of the previous works of the authors should be given. For example, I found at least one paper (Bioorg and Med Chem Lett, 2013, 23, 1091) from the same group of authors dealing with enzyme inhibition by structurally similar dihydropyrazole derivatives. Possibly, there are other published papers and a short overview of them would give the reader a better outlook of the work.
Answer: Thank you for your kindly recommendations. We have checked and revised manuscript to supplement our previous research work. The specific modifications are as follows:
Besides, the dihydropyrazole skeleton also appeared frequently in our previously published studies on COX-2 inhibitors such as A-16A, B-48 and C-4d.[30-32]
Figure 1. The design pathway for novel dihydropyrazole derivatives containing benzo oxygen heterocycle and sulfonamide moieties
[30]. Luo, Y.; Zhang, S.; Qiu, K. M.; Liu, Z. J.; Yang, Y. S.; Fu, J.; Zhong, W. Q.; Zhu, H. L., Synthesis, biological evaluation, 3D-QSAR studies of novel aryl-2H-pyrazole derivatives as telomerase inhibitors. Bioorg. Med. Chem. Lett., 2013, 23 (4), 1091-1095.
[31]. Chen, Z.; Wang, Z. C.; Yan, X. Q.; Wang, P. F.; Lu, X. Y.; Chen, L. W.; Zhu, H. L.; Zhang, H. W., Design, synthesis, biological evaluation and molecular modeling of dihydropyrazole sulfonamide derivatives as potential COX-1/COX-2 inhibitors. Bioorg. Med. Chem. Lett., 2015, 25 (9), 1947-1951.
[32]. Qiu, H. Y.; Wang, P. F.; Li, Z.; Ma, J. T.; Wang, X. M.; Yang, Y. H.; Zhu, H. L., Synthesis of dihydropyrazole sulphonamide derivatives that act as anti-cancer agents through COX-2 inhibition. Pharmacol. Res., 2016, 104, 86-96.
2. Compounds 2a-b were reported in the literature before, so the authors should give references for the preparation procedures that they used. Same goes for compounds 3a-z, it is unclear from the paper if some of these compounds were prepared for the first time. If that is the case, the authors should give full characterization for the new compounds. If all of these compounds were reported before, appropriate references should be given.
Answer: Thank you very much for your very good question and we are sorry for our carelessness. We have checked and supplemented the references for relevant reactions. The specific modifications are as follows:
3.2.1. General procedure for the synthesis of compound 2a-2b[33]
3.2.2. General procedure for the synthesis of compounds 3a-3z[34]
[33]. Pendergrass, K.; Hargreaves, R.; Petty, K. J.; Carides, A. D.; Evans, J. K.; Horgan, K. J., Aprepitant: an oral NK1 antagonist for the prevention of nausea and vomiting induced by highly emetogenic chemotherapy. RSC Advances, 2015, 5 (91), 74425-74437.
[34]. Taylor, A. M.; Schreiber, S. L., Aziridines as intermediates in diversity-oriented syntheses of alkaloids. Tetrahedron, 2017, 73 (24), 3368-3376.
3. CheckCIF report for structure 4b contains a lot of level A and level B alerts, which should be dealt with (or explanations for reasons why a specific alert could not be eliminated should be given) before this structure can be published.
Answer: Thank you for your kindly recommendations and we are sorry for our carelessness. We have checked and reprocessed single crystal data. The new 4b.CIF has been also uploaded together.
4. Furthermore, the overall quality of structure solution is not very good; some displacement ellipsoids are rather distorted, indicating that the disorder model used for structure solution can be improved.
Answer: Thank you for your kindly recommendations and we are sorry for our carelessness. We have checked and reprocessed single crystal data. The new 4b.CIF has been also uploaded together. The new data results are shown below:
Figure 2. Crystal structure diagrams of compound 4b
Table 2. Crystal data for compound 4b
Compound | 4b |
Empirical formula | C23H21N3O4S |
Formula weight | 435.5 |
Temperature (℃) | 161-162 |
Crystal system | Monoclinic |
Space group | P21 |
a (Å) | 5.3096(11) |
b (Å) | 19.638(4) |
c (Å) | 10.704(2) |
α (°) | 90.00 |
β (°) | 91.907(5) |
γ (°) | 90.00 |
V (Å) | 1115.9(4) |
Z | 2 |
Dcalcd/g cm-3 | 1.322 |
θ rang (deg) | 2.82-27.74 |
F (000) | 444 |
Reflections collected | 10926 (Rint= 0.0815) |
Data/restraints/parameters | 4715/1/273 |
Mu (mm-1) | 0.176 |
R1 | 0.1049 |
wR2 | 0.1385 |
GOOF | 1.118 |
Larg.peak/hole (e.Å) | 0.140/-0.469 |
CDCC number | 1900586 |
5. The resolution of images for NMR spectra given in the supplementary material is to low, which makes them almost useless. I suggest preparing a PDF document with vector images of NMR spectra.
Answer: We are sorry for our carelessness and thank you for your reminder. We have checked and remade it into a high-definition version of the PDF.

Reviewer 2 Report
Manuscript entitled “Dihydropyrazole Derivatives Containing Benzo Oxygen Heterocycle and Sulfonamide Moieties Selectively and Potently Inhibit COX-2: Design, Synthesis and Biological Evaluation” (molecules-471896) describes a series of sulfonamide derivatives designed to be COX-2 inhibitors. All new compounds 4a-z were identified based on ESI-MS, 1H NMR and elemental analysis. For compound 4b crystallographic structure was assigned.
The main claims of the paper concern the search for new COX-2 inhibitors as colon tumor therapeutics. The biological experiments including evaluation of cell viability (MTT test), COX-1/2 inhibition, apoptosis analysis and adhesion of cancer cells support the claims. Anticancer activity of 4b was also verified in SW620 xenograft mouse model. Moreover, molecular docking was applied to explain the binding mode as well as interactions of ligands and COX-2.
Reviewer recommends the paper to reconsideration after major revision that is required. An article has serious flaw.
First of all, the part “Introduction” needs serious correction. All references are inadequate to the text. For example, [1-2]- do not concern cyclooxygenases; [12]- does not concern cyclooxygenases; [13-14]- describe anti-inflammatory agents not anti-cancer drugs; [15]- does not concern overexpression of COX-2 in tumors; [16-17] – describe lipoxygenase inhibitors and COX-2 inhibitors as anti-inflammatory agents; [19] – tells about inhibition of adenosine kinase (AK); [20] – inhibition of lipid peroxidation; [21] – anti-inflammatory and antioxidant agents ; [19-21]- do not concern Celecoxib.
The title of the paper does not fully reflect the issues of presented work. The title should include anticancer activity.
Other main points:
1. Line 63; “In addition, cysteine in sulfonamides plays a crucial role in the selection of COX-2….” What does it mean that cysteine in sulfonamides plays role? Sulfonamide does not contain cysteine!
2. The graph in Fig 3 needs standard error bars.
3. In Fig 5, each group of tumor resection needs label (for example 2 days)
4. In Fig 6, it would be more interesting if authors balanced CDOCKER_INTERACTION_ENERGY against COX-2 inhibition potency.
5. Line 200; “As illustrated in Figure 7A and 7B, Celecoxib could effectively bind at this site through five..” – question is, which site?
6. Please, apply the IUPAC recommendations about 3-letter abbreviation for amino acid residues. For example, Arg, not ARG.
7. Line 214, should be DMSO-d6, not Dmso-6
8. Line 215, should be MHz, not mhz
9. Line 232, should be recrystallized from ethanol
10. All letters H, [d], p (in p-tolyl) in chemical names should be in Italic.
11. Line 239, should be “were refluxed”
12. Line 241, what does it mean, sodium sulfate was dried?
13. Please, add calculated molecular ion (MS analysis).
14. Line 228, remove "concentrated"
Author Response
Response to Reviewers
Dear Reviewers,
Many thanks for your comments. I really appreciate your kindly recommendations with our manuscript. According to the comments, we realize that our manuscript should be improved by a major revision. Therefore, we have revised our manuscript according to your comments. We hope the revised manuscript would be better. The main points of response to reviewers are as follow:
Reviewers' comments:
Reviewer #2: Manuscript entitled “Dihydropyrazole Derivatives Containing Benzo Oxygen Heterocycle and Sulfonamide Moieties Selectively and Potently Inhibit COX-2: Design, Synthesis and Biological Evaluation” (molecules-471896) describes a series of sulfonamide derivatives designed to be COX-2 inhibitors. All new compounds 4a-z were identified based on ESI-MS, 1H NMR and elemental analysis. For compound 4b crystallographic structure was assigned.
The main claims of the paper concern the search for new COX-2 inhibitors as colon tumor therapeutics. The biological experiments including evaluation of cell viability (MTT test), COX-1/2 inhibition, apoptosis analysis and adhesion of cancer cells support the claims. Anticancer activity of 4b was also verified in SW620 xenograft mouse model. Moreover, molecular docking was applied to explain the binding mode as well as interactions of ligands and COX-2.
First of all, the part “Introduction” needs serious correction. All references are inadequate to the text. For example, [1-2]- do not concern cyclooxygenases; [12]- does not concern cyclooxygenases; [13-14]- describe anti-inflammatory agents not anti-cancer drugs; [15]- does not concern overexpression of COX-2 in tumors; [16-17] – describe lipoxygenase inhibitors and COX-2 inhibitors as anti-inflammatory agents; [19] – tells about inhibition of adenosine kinase (AK); [20] – inhibition of lipid peroxidation; [21] – anti-inflammatory and antioxidant agents ; [19-21]- do not concern Celecoxib.
Answer: We are very sorry for our carelessness thank you very much for your circumspection. We have checked and made corresponding corrections. The specific modifications are as follows:
[1] Yasmin, S. K., Hugo G.; Lars, B.; Johan, Å., Toward an optimal docking and free energy calculation scheme in ligand design with application to COX-1 inhibitors, J. Chem. Inf. Model., 2014, 54 (5), 1488-1499.
[2] Peng, Q. L.; Yang, S.; Lao, X. J.; Tang, W. Z.; Chen, Z. P.; Lai, H.; Wang, J.; Sui, J. Z.; Qin, X.; Li, S., Meta-analysis of the association between COX-2 polymorphisms and risk of colorectal cancer based on case-control studies, PLoS One, 2014, 9 (4), e94790.
[12] Luisa, M. H., Cyclooxygenase-2 (COX-2) in inflammatory and degenerative brain diseases, J. Neuropathol. Exp. Neurol., 2004, 63 (9), 901-910.
[13] Amanda, K.; Jason, E. T.; Katherine. O.; Juliet, G. C.; Sara, Z.; Jordan, B.; Roderich, E. S.; Francis, J. B.; Rolf, A. B., Apricoxib, a novel inhibitor of COX-2, markedly improves standard therapy response in molecularly defined models of pancreatic cancer, Clin. Cancer Res., 2012, 18 (18), 5031-5042.
[14] Mohsen, V.; Mohsen, A., The discovery and development of cyclooxygenase-2 inhibitors as potential anticancer therapies, Expert. Opin. Drug Discov., 2014, 9 (3), 255-267.
[15] Zhang, H.; Sun, X. F., Overexpression of cyclooxygenase-2 correlates with advanced stages of colorectal cancer, Am. J. Gastroenterol., 2002, 97 (4), 1037-1041.
[16] Asako, O.; Asako, O.; Toshiro, S.; Toshinori, B.; Katsumi, S.; Katsuyuki, H.; Akinobu, G.; Masato, F.; Masato, K., a selective cyclooxygenase-2 inhibitor, induces upregulation of E-cadherin and has antitumor effect on human bladder cancer cells in vitro and in vivo, Urology, 2008, 71 (1), 156-160.
[17] Kimmie, N.; Jeffrey, A. M.; Andrew, T. C.; Kaori, S.; Jennifer, A. C.; Donna, N.; Leonard, B. S.; Robert, J. M.; Al, B. B.; Paul, L. S.; Renaud, W.; Alexander, H.; Richard, M. G.; Alan, P. V.; Shuji, O.; Edward, L. G.; Charles, S. F., Aspirin and COX-2 Inhibitor Use in Patients With Stage III Colon Cancer, J. Natl. Cancer Inst., 2015, 107 (1), dju345.
[19] Randall, E. H.; Galal, A. A.; Hussein, A.; Karen, S., Chemoprevention of Breast Cancer in Rats by Celecoxib, a Cyclooxygenase 2 Inhibitor, Clin. Cancer Res., 2000, 60 (8) 2101-2103.
[20] Verena, J., Targeting apoptosis pathways by Celecoxib in cancer, Cancer Lett., 2013, 332 (2) 313-344.
[21] Quiñones, O. G.; Pierre, M. B., Cutaneous Application of Celecoxib for Inflammatory and Cancer Diseases, Curr. Cancer Drug Targets, 2019, 19 (12), 5-6.
The title of the paper does not fully reflect the issues of presented work. The title should include anticancer activity.
Answer: Thank you for your patience and circumspection. We are sorry for our carelessness. The title of the paper has been revised as follow:
Dihydropyrazole Derivatives Containing Benzo Oxygen Heterocycle and Sulfonamide Moieties Selectively and Potently Inhibit COX-2: Design, Synthesis and Anti-Colon Cancer Activity Evaluation
Other main points:
1. Line 63; “In addition, cysteine in sulfonamides plays a crucial role in the selection of COX-2….” What does it mean that cysteine in sulfonamides plays role? Sulfonamide does not contain cysteine!
Answer: Thank you for your kindly recommendations and we are sorry for our carelessness.
In addition, para aminobenzene sulfonamide in sulfonamides plays a crucial role in the selection of COX-2, and many COX-2 inhibitors, such as Celecoxib, Valerian and Parecoxib, have para aminobenzene sulfonamide substitution on the aryl ring.
2. The graph in Fig 3 needs standard error bars.
Answer: We are sorry for our carelessness and thank you for your reminder. We have checked and reprocessed graph with standard deviation. The new Fig 3. is as follows:
3. In Fig 5, each group of tumor resection needs label (for example 2 days)
Answer: We are sorry for our carelessness and thank you for your reminder. We have checked and revised Fig 5 as follow:
4. In Fig 6, it would be more interesting if authors balanced CDOCKER_INTERACTION_ENERGY against COX-2 inhibition potency.
Answer: Thank you very much for your very good question. We have made a relationship fitting curve between CDOCKER_INTERACTION_ENERGY and COX-2 inhibition potency. The fitting results show that their relationship behaves approximately linear (y=46.82-0.86x, R2=0.72). The R value is approximately equal to 0.85 and approaches to 1, which not only indicates that the fitting curve has a good credibility, but also indicates that our design idea is reasonable.
The specific results are as follows:
Besides, linear fitting was performed by comparing the binding energy and COX-2 inhibition. It behaved approximate linearity (y=46.82-0.86x, R2=0.72) exhibited in the Figure7. The R value is approximately equal to 0.85 and approaches to 1, which not only indicates that the fitting curve has a good credibility, but also indicates that our design idea is reasonable.
Figure7. The fitting correlation between the binding energy and COX-2 inhibition
5. Line 200; “As illustrated in Figure 7A and 7B, Celecoxib could effectively bind at this site through five...” – question is, which site?
Answer: Thank you very much for your very good question. Known PDB 3LNI includes the Celecoxib binding site to the COX-2 protein. The analysis for PDB 3LN1 by DS 4.5 shows the XYZ axis data corresponding to the specific coordinates of binding site are 30.99, -22.28 and -16.51, respectively. The radius of the site is 6.96 Å.The specific modifications in the article are as follows:
As illustrated in Figure 8A and 8B, Celecoxib could effectively bind at this site (XYZ axis: 30.99, -22.28 and -16.51; radius: 6.96 Å) through five hydrogen bond receptors and eleven Pi bonds, with the stronger interactional amino acids for Arg-106, Ser-339, Arg-499, Gln-178 and Leu-338.
6. Please, apply the IUPAC recommendations about 3-letter abbreviation for amino acid residues. For example, Arg, not ARG.
Answer: Thank you for your patience and circumspection. We are sorry for our carelessness. The manuscript has been checked carefully and 3-letter abbreviation for amino acid residues errors have been revised.
7. Line 214, should be DMSO-d6, not Dmso-6
Answer: Thank you for your patience and circumspection. We are sorry for our carelessness. The sentences with relevant errors are modified as follows:
All 1H NMR spectra were recorded in DMSO-d6 with Bruker AM 600 MHz made by Rhenistetten Forchheim Company in Germany as the internal standard.
8. Line 215, should be MHz, not mhz
Answer: We are sorry for our carelessness and thank you for your circumspection. The specific modifications are as follows:
All 1H NMR spectra were recorded in DMSO-d6 with Bruker AM 600 MHz made by Rhenistetten Forchheim Company in Germany as the internal standard.
9. Line 232, should be recrystallized from ethanol
Answer: Thank you for your kindly recommendations and we are sorry for our carelessness. We have checked and revised the errors as follow:
A white precipitate (compound 2a-2b) was formed which was filtered and recrystallized from ethanol.
10. All letters H, [d], p (in p-tolyl) in chemical names should be in Italic.
Answer: We are sorry for our carelessness and thank you for your reminder. We have checked and made corresponding modifications.
11. Line 239, should be “were refluxed”
Answer: We are sorry for our carelessness and thank you for your reminder. We have checked and made corresponding modifications as follow:
The compounds 3a-3z (1 mmol), 4-sulfamethylhydrazine hydrochloride (1.2 mmol) and glacial acetic acid (0.5 mmol) were refluxed overnight in ethanol (15 mL).
12. Line 241, what does it mean, sodium sulfate was dried?
Answer: We are sorry for our carelessness and thank you for your circumspection. We have checked and made corresponding modifications as follow:
Saturated sodium bicarbonate was used to wash the organic anhydrous layer, which was then filtered to dry by anhydrous sodium sulfate and concentrated in vacuum.
13. Please, add calculated molecular ion (MS analysis).
Answer: Thank you very much for your favorable suggestion. We have checked and added calculated molecular ion.
14. Line 228, remove "concentrated"
Answer: Thank you for your kindly recommendations and we are sorry for our carelessness. We have checked the manuscript and remove superfluous words.

Round 2
Reviewer 1 Report
The authors have taken care of the reviewer's concerns and the manuscript can now be accepted
Author Response
The authors have taken care of the reviewer's concerns and the manuscript can now be acceptedAnswer: Many thanks to reviewer for your patient and careful recommendations.
Reviewer 2 Report
Authors included all comments from the reviewer.
Suggestion to new version of the paper is:
Line 62-“para aminobenzene sulfonamide” should be changed into “para-sulfamoylphenyl moiety”
Line 64- “para aminobenzene sulfonamide substitution on the aryl ring.” - please delete this fragment
Author Response
Response to Reviewers
Dear Reviewers,
Many thanks for your comments. I really appreciate your kindly recommendations with our manuscript. According to the comments, we realize that our manuscript should be improved by a minor revision. Therefore, we have revised our manuscript according to your comments. We hope the revised manuscript would be better. The main points of response to reviewers are as follow:
Reviewers' comments:
Reviewer #2:
Authors included all comments from the reviewer.
Suggestion to new version of the paper is:
Line 62-“para aminobenzene sulfonamide” should be changed into “para-sulfamoylphenyl moiety”
Line 64- “para aminobenzene sulfonamide substitution on the aryl ring.” - please delete this fragment
Answer: We are very sorry for our carelessness and thank you very much for your circumspection. We have checked and made corresponding corrections. The specific modifications are as follows:
In addition, para-sulfamoylphenyl moiety in sulfonamides plays a crucial role in the selection of COX-2, and many COX-2 inhibitors, such as Celecoxib, Valerian and Parecoxib.
This manuscript is a resubmission of an earlier submission. The following is a list of the peer review reports and author responses from that submission.